# Azelnidipine Exhibits In Vitro and In Vivo Antiviral Effects against Flavivirus Infections by Targeting the Viral RdRp

**DOI:** 10.3390/v14061228

**Published:** 2022-06-05

**Authors:** Zhuang Wang, Yunzheng Yan, Qingsong Dai, Yijie Xu, Jiye Yin, Wei Li, Yuexiang Li, Xiaotong Yang, Xiaojia Guo, Miaomiao Liu, Xingjuan Chen, Ruiyuan Cao, Wu Zhong

**Affiliations:** 1Institute of Medical Research, Northwestern Polytechnical University, Xi’an 710072, China; todd1997@mail.nwpu.edu.cn; 2National Engineering Research Center for the Emergency Drug, Beijing Institute of Pharmacology and Toxicology, Beijing 100850, China; yyzdlyk@163.com (Y.Y.); qingsong4321@126.com (Q.D.); beilou2353@163.com (Y.X.); yinjiye11@163.com (J.Y.); a_moon1096@163.com (W.L.); lyx1986528@126.com (Y.L.); yangxiao8792@163.com (X.Y.); 15227117791@163.com (X.G.); lmm200821@126.com (M.L.)

**Keywords:** azelnidipine, Zika virus, antiviral, flavivirus, RdRp

## Abstract

Flaviviruses, represented by Zika and dengue virus (ZIKV and DENV), are widely present around the world and cause various diseases with serious consequences. However, no antiviral drugs have been clinically approved for use against them. Azelnidipine (ALP) is a dihydropyridine calcium channel blocker and has been approved for use as an antihypertensive drug. In the present study, ALP was found to show potent anti-flavivirus activities in vitro and in vivo. ALP effectively prevented the cytopathic effect induced by ZIKV and DENV and inhibited the production of viral RNA and viral protein in a dose-dependent manner. Moreover, treatment with 0.3 mg/kg of ALP protected 88.89% of mice from lethal challenge. Furthermore, using the time-of-drug-addition assay, the enzymatic inhibition assay, the molecular docking, and the surface plasmon resonance assay, we revealed that ALP acted at the replication stage of the viral infection cycle by targeting the viral RNA-dependent RNA polymerase. These findings highlight the potential for the use of ALP as an antiviral agent to combat flavivirus infections.

## 1. Introduction

Flaviviruses, represented by dengue virus (DENV) and Zika virus (ZIKV), belong to the family *Flaviviridae*, a class of single-stranded, enveloped RNA viruses with hundreds of members, such as DENV, ZIKV, Japanese encephalitis virus (JEV), West Nile virus (WNV), and yellow fever virus (YFV). DENV is the most globally widespread arbovirus [1]. There are four antigenically distinct serotypes of DENV, and one person infected with DENV can also be reinfected with other serotypes [2]. Although Dengvaxia is currently available for DENV infection, it is not suitable for serotype-negative populations due to the antibody-dependent enhancement (ADE) of infection effect [3,4]. ZIKV is reportedly transmitted via *Aedes* mosquitoes, but other transmission routes, such as mother-to-child transmission and sexual contact, have also been reported [5]. Recent studies have shown that ZIKV can cause microcephaly in neonates and serious neurological complications in adults, such as Guillain–Barré syndrome, meningoencephalitis, myelitis, and epilepsy [6,7,8]. Flaviviruses are widespread and persistent worldwide, and have pandemic potential. There is an urgent need for specific antiviral drugs to combat flavivirus infections.

Dihydropyridine calcium channel blockers (DHP-CCBs), a series of compounds approved for the treatment of cerebrovascular diseases, have exhibited favorable efficacy against various virus infections [9]. Amlodipine exerted potent anti-Ebola virus activities in vivo [10]. Benidipine hydrochloride showed antiviral activity against severe fever with thrombocytopenia syndrome virus (SFTSV) in vitro and in vivo, impairing virus internalization and genome replication [11]. Another study demonstrated that manidipine blocked JEV/ZIKV infections by targeting the viral nonstructural (NS) protein NS4B, thus presenting a potential strategy for the treatment of flavivirus infections [12]. Azelnidipine (ALP), a classical third-generation DHP-CCB, has been approved as an antihypertensive drug [13]. Previous studies have revealed a number of pharmacological properties of ALP, including tumor immunity, oxidative stress, and inflammatory response [14,15,16]. However, the anti-flavivirus potential of ALP remains to be elucidated.

In our study, ALP exhibited potent antiviral activity against ZIKV and DENV in vitro and in vivo. Mode-of-action studies have revealed that ALP directly targets the viral RNA-dependent RNA polymerase (RdRp) and inhibits viral replication. These findings should help in constructing a novel antiviral approach with DHP-CCB distinct from that described in previous reports, and highlights the potential for the use of ALP as an anti-flavivirus drug candidate.

## 2. Materials and Methods

### 2.1. Cells, Viruses, and Reagents

Vero (Cat #CCL-81) cells were purchased from the American Type Culture Collection (ATCC, Bethesda, MD, USA). BHK cells were purchased from the National Infrastructure of Cell Line Resource, China. Both cell lines were cultured in Dulbecco’s modified Eagle’s medium (DMEM, Gibco, Grand Island, USA) supplemented with 10% fetal bovine serum (FBS, Gibco, Grand Island, USA) and 1% penicillin/streptomycin at 37 °C and in 5% CO_2_. Flaviviruses including ZIKV (SMGC-1 strain) and DENV (serotype 2, New Guinea C (NGC) strain) were passaged and propagated as previously described [17]. The virus titers were determined by plaque-forming unit (PFU) assays or 50% tissue culture infective dose infectivity assays (TCID_50_).

Azelnidipine (MCE, Monmouth Junction, NJ, USA, cat #HY-B0023), 7-deaza-2′-C-acetylene-adenosine (NITD008; MCE, Monmouth Junction, NJ, USA, cat #HY-12957), 2′-C-methyladenosine (2′-CMA; TargetMol, Boston, MA, USA, cat #T16325), and 3′-dATP (Sigma, St. Louis, MO, USA, cat # C9137) were dissolved in DMSO and stored at −20 °C until used.

### 2.2. In Vitro Antiviral Assay

BHK and Vero cells were used for the evaluation of ALP’s antiviral effects against DENV and ZIKV, respectively. The cytopathic effect (CPE) protection assay was performed according to previously reported methods [18]. Briefly, BHK or Vero cells were seeded in 96-well plates at a density of 5 × 10^3^ or 1 × 10^4^ cells per well, and incubated at 37 °C overnight. The cells were treated with threefold serially diluted ALP and 100 × TCID_50_ viruses until an obvious CPE was observed. The cell viability of each group was then tested in accordance with the procedure of the CellTiter-Glo Luminescent Cell Viability kit. The half-maximal inhibitory concentration (IC_50_) value of ALP was fitted and calculated using the Origin 9.0 software. The assessment method for the 50% cytotoxic concentration (CC_50_) was similar to that for IC_50_, except that the virus inoculum was replaced with DMEM containing 2% FBS.

For the virus yield reduction assay, the BHK or Vero cells were seeded in a 12-well plate and cultured overnight, and they were then inoculated with gradient-diluted ALP in the presence of the viruses (multiplicity of infection (MOI) = 0.05). After 72 h of incubation, the cellular RNA was extracted using the RNeasy^®^ Mini Kit and quantified using qRT-PCR, as previously reported [19]. The viral proteins were visualized using an immunofluorescence assay. Briefly, the BHK or Vero cells seeded in the 96-well plate were inoculated with virus inoculum at an MOI of 0.05 PFU/mL. Meanwhile, the diluted compound or DMSO was added. The cells were fixed with 4% paraformaldehyde at 48 h post-infection (h.p.i.) and then treated with 0.1% Triton X-100 to allow cellular permeabilization, followed by blocking with 5% BSA at room temperature for 1 h. The primary antibody for the mouse anti-flavivirus group antigen (1:500) (Merck Millipore, Darmstadt, Germany, cat #MAB10216) and the corresponding secondary antibody (Invitrogen, Carlsbad, CA, USA, cat #A32727) were diluted and incubated with the cells in turn. Then, the cells were washed and stained with Hoechst 33342 fluorescent stain (Thermo Fisher Scientific, Waltham, MA, USA, cat #H21492), and images were taken using a Leica DMi8.

### 2.3. Time-of-Drug-Addition Assay

A time-of-drug-addition assay was performed as previously described with some modifications [17,18]. Briefly, BHK cells were seeded at a density of 2.5 × 10^5^ cells/well in a 12-well plate and cultured at 37 °C overnight. The cells were inoculated with ZIKV or DENV for 0 to 2 h, and ALP (6 μM) or the control drug NITD008 (5 μM) was added at four intervals, which shows different stages of the virus infection cycle. The cells in each group were washed three times with DMEM containing 2% FBS after the treatment. The intracellular viral RNA was extracted at 24 h.p.i. and determined using the qRT-PCR method.

### 2.4. ZIKV Replicon Inhibition Assay

The ZIKV replicon inhibition assay was carried out as described previously [18]. Confluent BHK–ZIKV replicon cells were co-cultured with the gradient-diluted compounds at 37 °C for 48 h. After that, the cells were washed with PBS and assessed using the Dual-Luciferase Reporter Assay System kit. The inhibitor 2′-CMA was used as the positive control.

### 2.5. Quantitative Real-Time PCR (qRT-PCR)

The total cellular RNA was harvested, and the viral RNA was quantified using qRT-PCR according to the procedure of the One Step PrimeScript RT-PCR kit. The specific primers and probes used for ZIKV and DENV assessment were as follows: ZIKV forward primer—GGTCAGCGTCCTCTCTAATAAACG;Reverse primer—GCACCCTAGTGTCCACTTTTTCC;Probe—AGCCATGACCGACACCACACCGT;DENV forward primer—AGGTCGGATTAAGCCATAGTACG;Reverse primer—TGGCCTGACTTCTTTTAACGTC;Probe—AAAAACTATGCTACCTGTGAGCCCCGTCC.

The viral RNA copies were determined from the cycle threshold (Ct) value of each sample, with reference to the known copy number standard curve.

### 2.6. RdRp Enzymatic Inhibition Assay

The DENV RdRp protein was expressed and purified, and a real-time fluorescence-based RdRp enzymatic system was constructed and optimized as described previously [20,21]. To evaluate the inhibitory activity of ALP against the RdRp enzyme, the target proteins (500 nM) were pre-incubated for 15 min with gradient-diluted ALP at final concentrations ranging from 50 to 3.125 μM. After that, the mixture of the other reaction components, including ATP, poly U, and SYTO 9, was added into the 96-well black plate to initiate the reaction. The total volume of the reaction system was 100 μL. The changes in real-time fluorescence intensity (Ex. 485 nm, Em. 520 nm) in each well were monitored and recorded every 30 s for 60 min using an EnSpire multifunctional microplate reader. The nucleotide 3′-dATP was used as the positive control. The normalized data were then used to calculate the IC_50_ values of the test compounds using the Origin 9.0 software.

### 2.7. Surface Plasmon Resonance (SPR) Assay

The SPR assay was carried out using the Biacore 8K (Cytiva, Uppsala, Sweden) instrument to explore the interaction of DENV-RdRp protein with ALP. RdRp proteins in 10 mM sodium acetate were immobilized on the CM5 chip. Then, gradient-diluted ALP (50–3.13 μM) in a running buffer consisting of 1 × PBS with 0.05% Tween-20 and 5% DMSO at pH 7.4 flowed over the sensor chip surface. The response units were recorded and fitted using the Biacore Evaluation Software.

### 2.8. Molecular Docking Simulations

The crystal structures of DENV RdRp protein (PDB: 5K5M) and ZIKV RdRp protein (PDB: 6LD5) were obtained from the Protein Data Bank (PDB) database. The “Prepare Protein” module of the Discovery Studio 4.5 software was used to remove the conformation of the nontarget protein and supplement the incomplete amino acid residues. ALP was prepared using the “Prepare Ligands” module to obtain an effective three-dimensional conformation, and then docked with CDOCKER for target proteins. The key interactions and amino acid residues were analyzed using the PyMOL 2.5 software.

### 2.9. In Vivo Antiviral Efficacy

The animals used in this study were specific pathogen-free grade, and all the experiments and operations were reviewed and approved by the Institutional Animal Care and Use Committee of Beijing Institute of Pharmacology and Toxicology. One-day-old ICR suckling mice were used to evaluate the in vivo anti-ZIKV efficacy of ALP. Briefly, each suckling mouse was intraperitoneally (i.p.) injected with ZIKV at a volume of 25 μL containing 1.3 × 10^4^ PFU of virus particles, followed by the administration of ALP (0.3 mg/kg, 0.03 mg/kg) or the vehicle at 4 h post-infection. After that, the mice in each group were i.p. treated once daily for 9 consecutive days. The survival and bodyweight of the infected mice were recorded for 21 days.

### 2.10. Statistical Analyses

Statistical analyses were performed using the GraphPad Prism 7 software. An unpaired two-tailed Student’s *t*-test or one-way analysis of variance was used for data analysis. The survival curve was determined using the log-rank test. The results are expressed as the means ± standard error of mean (SEMs). Considered statistically significant was *p* < 0.05.

## 3. Results

### 3.1. ALP Exhibits Favorable Antiviral Effects against ZIKV and DENV Infections In Vitro

To explore the in vitro anti-flavivirus potential of ALP (Figure 1A), two representative viral species, ZIKV and DENV, were employed. A CPE inhibition assay was performed to evaluate the effects of ALP against flavivirus infections. As shown in Figure 1D,E, ALP exhibited potent antiviral activity (without cytotoxicity) in a dose-dependent manner; the IC_50_ values were 1.68 ± 0.50 μM and 0.76 ± 0.17 μM for ZIKV and DENV, respectively. Then, we performed qRT-PCR and immunofluorescence assays to assess the effects of ALP on the yields of viral RNA and proteins. As shown in Figure 1B and C, ALP inhibited the production of viral RNA in a dose-dependent fashion. Similarly, ALP significantly reduced the expression of the viral envelope protein at a concentration of 5 μM (Figure 1F,G). The data suggested the in vitro antiviral potential of ALP against flavivirus infections.

### 3.2. ALP Mainly Acts at the Post-Entry Stage of Flavivirus Infections

To determine the mechanism of action of ALP, a time-of-drug-addition assay was first conducted to determine the specific stage at which ALP acted, as shown in the schematic illustration in Figure 2A. As shown in Figure 2B, ALP and NITD008 (the positive control drug in the viral replication stage) treatment significantly decreased intracellular viral RNA production in stages III and IV, which suggests that ALP mainly acted at the post-entry stage of the ZIKV infection cycle. Similarity, we observed that the replication stage of DENV was potently inhibited by ALP (Figure 2C). Furthermore, we performed a ZIKV replicon assay using 2′-CMA as the positive control drug to further determine whether ALP interfered with intracellular viral replication. As shown in Figure 2D, ALP and 2′-CMA exhibited comparable inhibitory effects, with IC_50_ values of 1.67 ± 0.01 μM and 2.70 ± 0.06 μM, respectively. Collectively, these data suggest that ALP mainly acted at the post-entry stage of flavivirus infection.

### 3.3. ALP Exerts Antiviral Effects by Inhibiting the Viral RNA-Dependent RNA Polymerase

RNA-dependent RNA polymerase (RdRp) is the key viral protein with enzymatic activity and is responsible for the replication of the viral genome. Considering that the RdRp protein is a key enzyme in the viral replication process, we hypothesized that ALP might exert its antiviral effects through the inhibition of viral RdRp protein. To this end, a real-time fluorescence-based enzymatic system using the DENV RdRp protein was first constructed and optimized according to previously reported methods [21]. We then evaluated the inhibitory activity of ALP against the RdRp enzyme. As shown in Figure 3A, the fluorescence intensity for the ALP-treated group increased more slowly than that for the mock group, and the signal values for the high-ALP-concentration group (50 and 25 μM) were hardly increased. Furthermore, the IC_50_ values of ALP and the positive control compound 3′-dATP were determined to be 6.72 ± 4.69 μM and 30.09 ± 8.26 μM, respectively (Figure 3B). To further verify the direct interaction of ALP with the RdRp protein, an SPR assay was performed. As shown in Figure 3C, the response units increased in a dose-dependent manner with an increase in the ALP concentration, and the *K_D_* value was 73.7 μM. Taken together, all these data show that ALP exhibited anti-flavivirus activity by targeting the viral RdRp protein.

### 3.4. ALP Binds to the N Pocket of RdRp Protein In Silico

To further explore the mode of ALP’s binding with the flavivirus RdRp protein, an *in silico* ligand-docking assay was performed using the solved crystal structure of the flavivirus RdRp protein. The RdRp protein consisted of three subdomains, namely, the palm, thumb, and fingers subdomains. To date, two modes of small molecular inhibitors’ binding with the flavivirus RdRp protein have been reported [22,23,24]. First, the nucleoside analogs NITD008 and 7DMA act on the catalytically active sites via incorporation into the synthesizing RNA strand to terminate chain elongation. Second, some inhibitors target the allosteric pocket, termed the N pocket, and prevent conformational changes in the RdRp protein. Here, the results of molecular-docking assays using ZIKV RdRp protein (PDB: 6LD5) and DENV RdRp protein (PDB: 5K5M) indicated that ALP bound to the N pocket by anchoring on the priming loop (Figure 4A–D). Notably, the nitrobenzene group of ALP formed a cation–π interaction with Arg729 in the DENV RdRp protein (PDB: 5K5M), while two ester groups formed hydrogen bonds with the side chains of Arg729, Arg737, Thr794, Trp795, and Ser796. The diphenylmethyl group of ALP filled the hydrophobic pocket formed by Trp795, Phe349, Ile474, and Tyr476. When docked with ZIKV RdRp protein (PDB: 6LD5), we observed that the nitro group of ALP formed hydrogen bonds and salt bridges with the positively charged residues Arg731 and Lys462, a process similar to the binding modes of previously reported inhibitors [25,26]. The diphenylmethyl group of ALP occupied a similar hydrophobic pocket (Figure 4E,F). In summary, we can speculate that ALP may exhibit anti-flavivirus activities by interacting with the priming loop, thereby preventing the conformational change in the flavivirus RdRp protein.

### 3.5. ALP Exerts Antiviral Efficacy against ZIKV Infection In Vivo

To evaluate the in vivo antiviral effects of ALP, we first utilized the ZIKV lethal infection model to evaluate the anti-ZIKV potential of the tested compound. One-day-old ICR suckling mice were inoculated with 1.2 × 10^4^ PFU per mouse of ZIKV. Mice in different groups were administered the vehicle or ALP at a dose of 0.03 or 0.3 mg/kg at 4 h post-infection, followed by the same dosage once daily for 9 consecutive days. As shown in Figure 5A,B, the vehicle group all died within 16 days, whereas the survival rate of the treatment group significantly improved. The 0.3 mg/kg treatment group reached 88.9%. Meanwhile, the bodyweights of the ALP-treated group steadily increased throughout the experiment.

To further determine the protective effect of ALP, we euthanized and dissected the mice on the third day after infection and examined the viral load in different tissues. The results show that, after challenge, the ALP-treated group showed significantly reduced viral loads in the brain and liver compared with the untreated control mice. However, no significant difference of viral loads in the spleen between the control and the treatment was observed (Figure 5C). Due to the neurotropic nature of ZIKV, we performed H&E staining on brain tissue (Figure 5D). In the mock group, we observed inflammatory cell infiltration and neuronal cell necrosis caused by viral infection. ALP treatment alleviated the symptoms and decreased inflammatory cells. We also performed immunohistochemical assays using specific anti-ZIKV-NS2B antibodies (GeneTex, Irvine, CA, USA, cat #GTX133308) to visualize the virus particles in the stained images (Figure 5D). As shown, relatively few viral antigens were detected in the ALP-treated group. All the above data further confirm the antiviral efficacy of ALP against ZIKV infection in vivo.

## 4. Discussion

Flaviviruses transmitted through multiple routes could lead to the development of a variety of diseases with serious consequences [1]. However, no specific anti-flavivirus drugs have been approved yet. In the present study, we found that the FDA-approved drug ALP exhibited potent efficacy against ZIKV and DENV infections in the CPE protection assays with IC_50_ values of 1.68 ± 0.50 μM and 0.76 ± 0.17 μM, respectively. Additionally, the yields of the viral RNA and proteins in the ALP-treated groups were decreased in a dose-dependent manner. To explore the mechanism of ALP’s action against ZIKV and DENV, we first carried out a time-of-addition assay, and the results revealed that ALP acted at the post-entry stage of flavivirus infection, which is consistent with the results obtained via the replicon inhibition assay. Considering that the RdRp protein plays a key role in intracellular viral replication, we constructed a real-time fluorescence-based enzymatic system and evaluated the effects of ALP against the viral RdRp protein. To our surprise, a potent inhibitory effect (IC_50_ = 6.72 ± 4.69 μM) was observed in the RdRp catalytic assay in accordance with the results obtained in the in vitro antiviral assays and the SPR assay. Furthermore, the results of the ligand-docking assay indicated that ALP may bind at an allosteric site anchored by the priming loop, specifically the N pocket of the flavivirus RdRp protein. 

It has been demonstrated that host cell dysfunction following viral infection is accompanied by abnormal intracellular Ca^2+^ concentrations [27]. One research group observed that WNV infection resulted in rapid and sustained Ca^2+^ influx using different WNV strains and different cell types. Treatment with BAPTA or EGTA, an extracellular calcium chelator, reduced the Ca^2+^ influx and decreased the viral yield early in infection. They suggest that WNV infection leads to the early persistent activation of FAK, ERK1/2, and PI3K/Akt in mammalian cells [28]. However, this was not observed in BAPTA-AM-treated, WNV-infected cells, suggesting that kinase activation requires viral replication and is mediated by Ca^2+^ signaling [29]. Wang et al. showed that the NS4B of flavivirus is a viral target of manidipine. NS4B Q130, substituted by a basic amino acid mutation, may be involved in the interaction of NS4B with host proteins, rather than viral proteins. This group also highlighted the importance of intracellular calcium to flaviviruses [12]. Interestingly, a recent study found that the calcium-channel inhibitor lacidipine inhibited the ZIKV independently of calcium. A plausible explanation is that its antiviral effect is related to the alteration of the transport of free cholesterol and neutral lipids [30]. In this study, we mainly explored the anti-flavivirus effects of ALP targeting the viral RdRp enzyme. The mechanism has not been explored in depth, by which ALP acts as a calcium channel blocker to inhibit the virus using host calcium channels. We also cannot deny the possibility that ALP may act on other host targets.

At present, there have been no reports on the use of azelnidipine for the treatment of infectious diseases. However, benidipine and nifedipine, two calcium channel blockers, inhibited SFTSV replication in vitro [11]. A retrospective clinical investigation on a large cohort of SFTS patients was also performed, and the results demonstrated that nifedipine administration enhanced virus clearance, improved clinical recovery, and remarkably reduced the case fatality rate. This may imply good prospects in the clinical application of calcium channel inhibitors. In our study, we have revealed that ALP has a potent inhibitory effect against flavivirus infection. Our data suggest that ALP is an attractive anti-flavivirus drug candidate that acts by targeting the viral RdRp protein, and ALP has potential for use as a promising novel skeleton lead for future drug design. Taken together, our findings will be highly valuable for the development of potential therapeutics for flaviviruses.

## Figures and Tables

**Figure 1 viruses-14-01228-f001:**
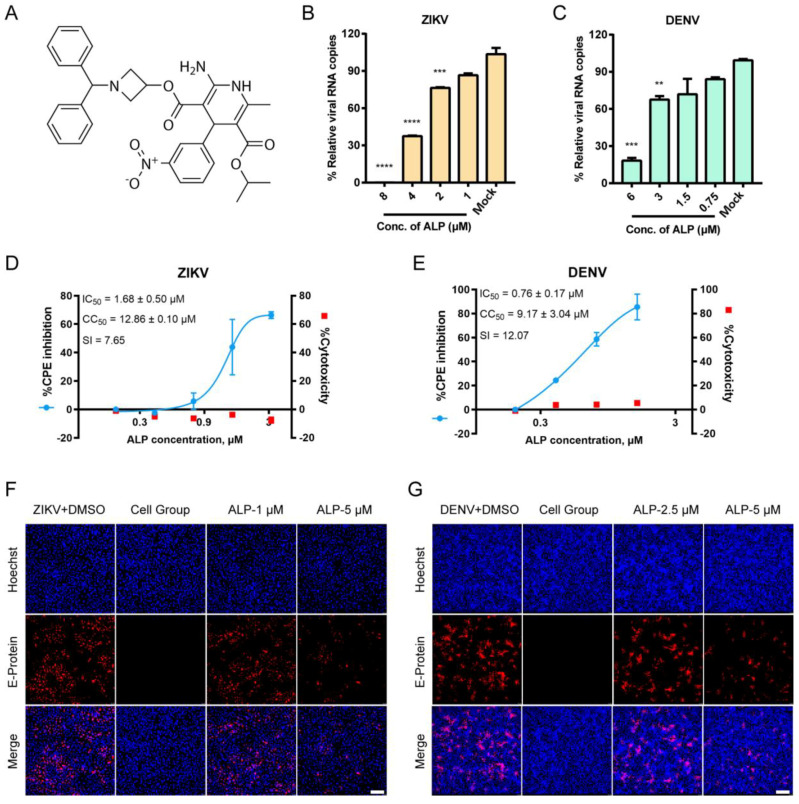
In vitro antiviral activities of Azelnidipine (ALP) against Zika and dengue virus (ZIKV and DENV). (**A**) Molecular structure of ALP. (**B**,**C**) Cells were cultured with ZIKV or DENV in the presence of different concentrations of ALP. Cellular RNA was extracted at 72 h post-infection (h.p.i.) and quantified using qRT-PCR to determine the viral RNA level. Data were collected from three independent experiments and analyzed using one-way analysis of variance. ** *p* < 0.01, *** *p* < 0.001, and **** *p* < 0.0001. (**D**,**E**) The cytopathic effect (CPE) protection efficacy of ALP against ZIKV or DENV was evaluated in Vero or BHK cell lines, respectively. The left and right y-axes represent the mean % CPE inhibition and cytotoxicity of the drug, respectively. SI (selectivity index) = CC_50_/IC_50_. (**F**,**G**) Fluorescence imaging of ALP against ZIKV and DENV. Cells were stained with Hoechst (blue) and the anti-flavivirus envelope protein (red). Scale bar: 200 µm.

**Figure 2 viruses-14-01228-f002:**
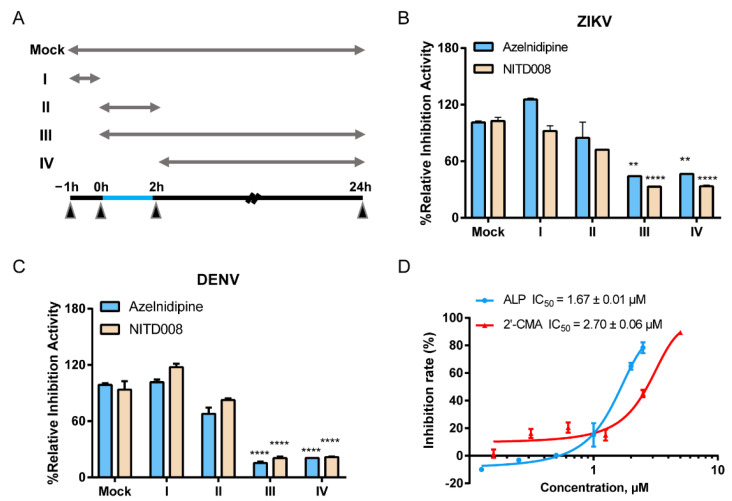
Time-of-drug-addition assay and the replicon assay for ALP. (**A**) Experimental design of the time-of-drug-addition assay. Virus diluent (blue line) was incubated with the cells at 0–2 h. ALP and the positive control drug NITD008 were added to the cells before, during, after, or throughout the whole infection, respectively, as indicated in the horizontal upper lines. The cellular RNA was extracted at 24 h.p.i. and quantitated by qRT-PCR. One-way analysis of variance (**B**,**C**) was performed for statistical analysis. Data are expressed as means ± standard error of mean. ** *p* < 0.01 and **** *p* < 0.0001. (**D**) The inhibitory effect of ALP against ZIKV replicon. Cells were treated with ALP at the indicated concentrations, and luciferase activities were measured at 48 h.p.i. The inhibitor 2′-CMA was set as the positive control. The inhibition curve was fitted using GraphPad Prism 7 software.

**Figure 3 viruses-14-01228-f003:**
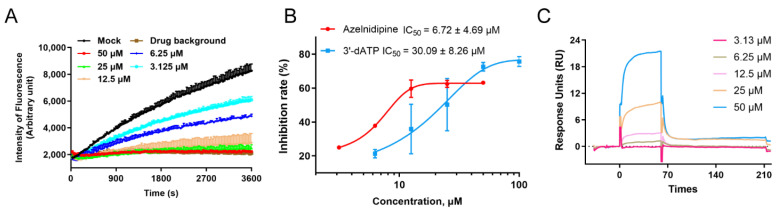
Enzymatic inhibition evaluation of ALP against flavivirus RdRp protein. (**A**) Real-time fluorescence changes in DENV RdRp enzymatic system inhibited by different concentrations of ALP (50–3.125 μΜ). (**B**) Dose–response inhibitory curve for ALP’s activity against the DENV RdRp protein. The nucleotide 3′-dATP was used as the positive control. (**C**) Surface plasmon resonance assay was conducted to determine the direct interactions between ALP and DENV RdRp protein. Immobilized DENV RdRp protein was coated onto a CM5 chip and incubated with gradient-diluted ALP for response unit monitoring. Representative results are as shown.

**Figure 4 viruses-14-01228-f004:**
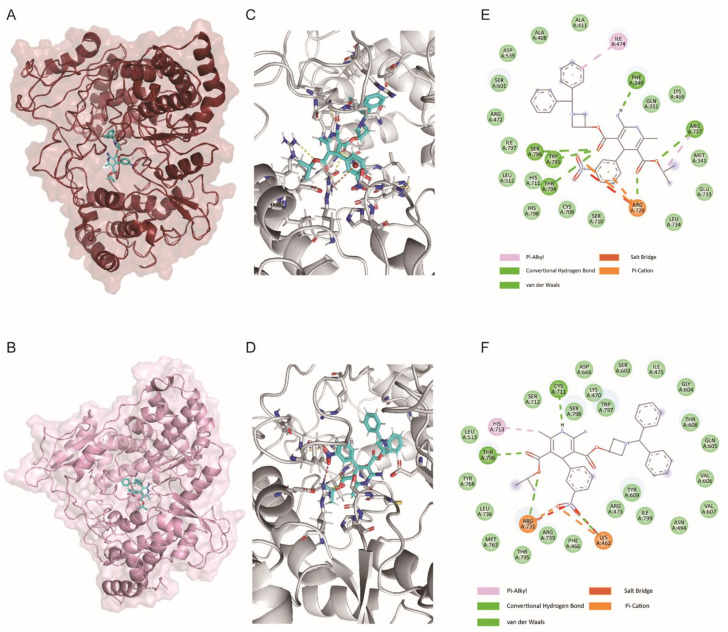
Predicted binding sites for ALP bound to DENV RdRp protein (**A**,**C**,**E**) or ZIKV RdRp protein (**B**,**D**,**F**). RdRp proteins of DENV (5K5M) and ZIKV (6LD5) were used for the ligand-docking assay. The overview and specific diagrams of the binding pose between ALP and RdRp protein are shown in the left panel (**A**,**B**) and middle panel (**C**,**D**), respectively. The predicted results for interacting residues and bonds are shown in right panel (**E**,**F**).

**Figure 5 viruses-14-01228-f005:**
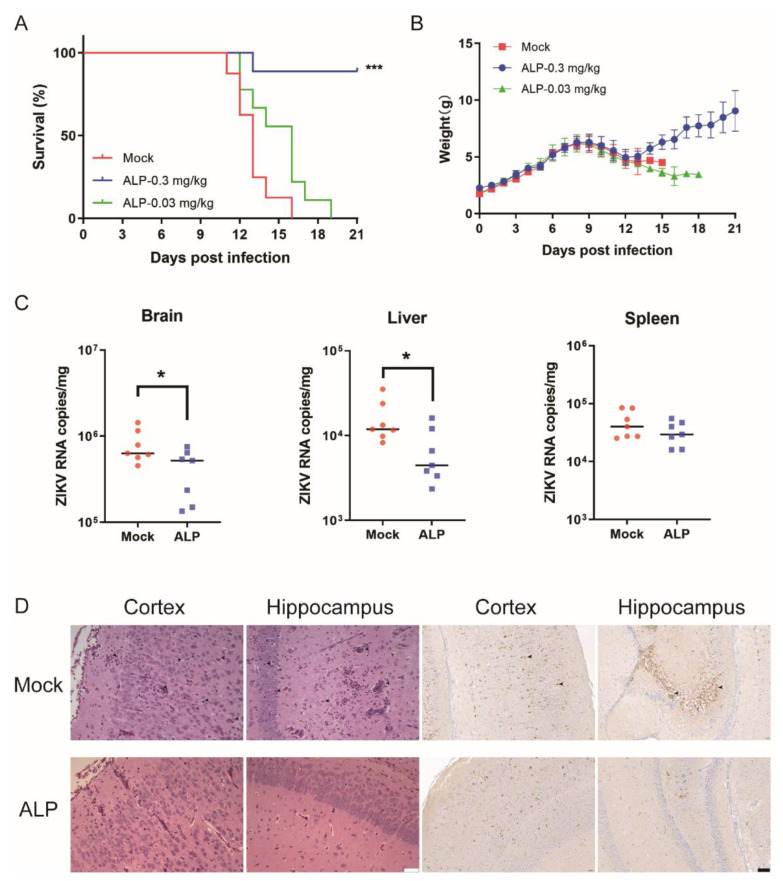
In vivo anti-ZIKV efficacy of ALP in the ZIKV lethal challenge mouse model. One-day-old ICR suckling mice were treated with 0.3 mg/kg or 0.03 mg/kg of ALP intraperitoneally (i.p.) 4 h after ZIKV-SMGC-1 challenge (1.2 × 10^4^ PFU/mouse, i.p.), and this was administered for 9 consecutive days. The changes in survival (**A**) and bodyweight (**B**) of the ZIKV-infected mice in each group were recorded until 21 d.p.i.. Survival data were analyzed using a log-rank test, *** *p* < 0.001. (**C**) The viral loads in mouse brains, livers, and spleens were measured by qRT-PCR. * *p* < 0.1. (**D**) H&E staining (scale bar = 50 µm) and immunohistochemistry assay (scale bar = 200 µm) of sectioned brains. At 3 days post-infection, azelnidipine treatment alleviated the histopathological changes in mice caused by ZIKV infection. The viral antigens were visualized using NS2B antibodies via an immunohistochemistry assay. Yellow staining is considered to show viral antigens.

## Data Availability

All data are available in the main text.

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
