# Peer review of "Azelnidipine Exhibits In Vitro and In Vivo Antiviral Effects against Flavivirus Infections by Targeting the Viral RdRp"

_viruses, 2022, doi:10.3390/v14061228_

Round 1

Reviewer 1 Report

The manuscript ‘Azelnidipine exhibits in vitro and in vivo antiviral effects against flavivirus infections by targeting the viral RdRp’ by Zhuang Wang et al. found azelnidipine (ALP) show potent anti-DENV and ZIKV activities in vitro and protected 88.89% of mice from ZIKV lethal challenge. Furthermore, the target of ALP was identified as the viral RNA-dependent RNA polymerase (RdRp). The logic of this study is sound by a progressive layers of the investigation process. ALP inhibited the infection of DENV and ZIKV in vitro. Then, ALP was found to disturb the post-entry stage of DENV and ZIKV infection cycle. And, the RdRp was idenfied as the target of ALP by an enzymatic inhibition assay, molecular docking and a surface plasmon resonance assay. Finally, the anti-ZIKV efficacy of ALP was evaluated with lethal mice model. The experimental evidence is conclusive to support the conclusion. As mentioned in the manuscript, dihydropyridine calcium-channel blockers (DHP-CCBs), a series of compounds approved for the treatment of cerebrovascular diseases, have been reported to exhibited favorable efficacy against various virus infections. And, the Ca2+ concentration of host cells is important for various viruses. ALP as the classical third-generation DHP-CCB, may play antiviral role through calcium-channel or other targets. So, the manuscript is relatively less innovative. But, the manuscript is logically clear, informative and have certain scientific significance, suggested to be accepted after minor revision.

  1. Figure1 d and 1E, the experiments to calculate IC50 of ALP anti-DENV or anti-ZIKV just including only 3-4 drug concentrations, the IC50 values may not be accurate. More drug concentrations are suggested.
  2. Line 182, there is a formatting error in the text, “was mentioned”.

Author Response

Point 1: Figure1 d and 1E, the experiments to calculate IC50 of ALP anti-DENV or anti-ZIKV just including only 3-4 drug concentrations, the IC50 values may not be accurate. More drug concentrations are suggested.

Response 1: We are grateful that this comment is very important to improve the quality of the manuscript. We have added more concentration gradients to Figure 1D-1E based on previous experimental data. Furthermore, we would like to further elaborate that we measured the IC50 values of ALP at non-toxic concentrations. Therefore, we present the IC50 fitted curve as shown. (Page 5, Fig 1D-1E).

Point 2: Line 182, there is a formatting error in the text, “was mentioned”.

Response 2: Thank you for pointing this mistake out. We have amended the related text (Page 4, Line 182).

Reviewer 2 Report

This is an interesting manuscript describing a study to evaluate anti-Flavivirus effects of azelnidipine (ALP) against Dengue virus (DENV) and Zika virus (ZIKV). ALP is an approved antihypertensive drug that with a mechanism of action involving blocking calcium-channels. The observations described here include prevention of the cytopathic effects, dose-dependent inhibition of viral RNA and viral proteins, and in vivo protective effects using an established mouse model. The authors provide convincing data showing that ALP targets the viral RNA-dependent RNA polymerase and effects the replication stage of viral infection, although they acknowledge that other host targets could also be involved. The molecular docking studies reveal a potential binding mode for ALP with the priming loop of the RdRp protein as a possible mechanism for the anti-flavivirus activity. The manuscript is organized and well-written with few apparent grammatical issues. Minor edits could correct a misspelling in line 160 “sulking”; and some clarification of the wording in the sentence (line 280) describing the results in the spleen where no significant differences between the control and treatment were observed would be appropriate.This article should be of interest to readers of this Journal, and these results may stimulate further study of the antiviral effect of this class of compound on flaviviruses.

Author Response

Point 1: Minor edits could correct a misspelling in line 160 “sulking”.

Response 1: Thank you for pointing this mistake out. We have corrected "sulking" to "suckling mouse" in the text (Page 4, Line 160).

Point 2:Some clarification of the wording in the sentence (line 280) describing the results in the spleen where no significant differences between the control and treatment were observed would be appropriate.

Response 2: Thanks a lot for your suggestion. We have revised the description in our latest submission (Page 10, Line 281).
